# Distribution of Foliicolous Lichen *Strigula* and Genetic Structure of *S. multiformis* on Jeju Island, South Korea

**DOI:** 10.3390/microorganisms7100430

**Published:** 2019-10-10

**Authors:** Seung-Yoon Oh, Jung-Jae Woo, Jae-Seoun Hur

**Affiliations:** 1Korean Lichen Research Institute, Sunchon National University, 255 Jungang-Ro, Suncheon 57922, Korea; syoh@snu.ac.kr (S.-Y.O.); lichenwoojae@korea.kr (J.-J.W.); 2Division of Forest Biodiversity, Korea National Arboretum, 415 Gwangneungsumok-ro, Pocheon 11186, Korea

**Keywords:** Dothideomycetes, foliicolous lichen, gene flow, Jeju Island, lichens, MaxEnt, population genetics, species distribution modeling, *Strigula multiformis*

## Abstract

*Strigula* is a pantropic foliicolous lichen living on the leaf surfaces of evergreen broadleaf plants. In South Korea, *Strigula* is the only genus of foliicolous lichen recorded from Jeju Island. Several *Strigula* species have been recorded, but the ecology of *Strigula* in South Korea has been largely unexplored. This study examined the distribution and genetic structure of *Strigula* on Jeju Island. The distribution was surveyed and the influence of environmental factors (e.g., elevation, forest availability, and bioclimate) on the distribution was analyzed using a species distribution modeling analysis. In addition, the genetic variations and differentiation of *Strigula*
*multiformis* populations were analyzed using two nuclear ribosomal regions. The distribution of *Strigula* was largely restricted to a small portion of forest on Jeju Island, and the forest availability was the most important factor in the prediction of potential habitats. The genetic diversity and differentiation of the *S. multiformis* population were found to be high and were divided according to geography. On the other hand, geographic and environmental distance did not explain the population differentiation. Distribution and population genetic analysis suggested that the available habitat and genetic exchange of *Strigula* on Jeju Island are limited by the lack of available forest in the lowlands.

## 1. Introduction

Foliicolous lichens are an ecological group of lichens living on the surface of plant leaves [1]. More than 800 species of foliicolous lichens have been found worldwide. They are distributed generally in subtropical and tropical forests and play important ecological roles in forest ecosystems [2]. The distribution and genetic variations of foliicolous lichens have received less attention than other lichen groups, likely due to their small and inconspicuous morphology [3,4]. Various environmental factors, such as climate, elevation, and vegetation are associated with the lichen distribution [5,6,7,8,9]. Therefore, an understanding of the relationship between environmental factors and lichen occurrence is key to identifying and predicting the distribution of lichens [10]. In addition, the genetic variations of the lichen population are important because they are a source of environmental adaptation [11]. A loss of genetic variation reduces the adaptive potential against environmental stress, such as climate change and habitat disturbance [12,13]. Similar to the distribution pattern, the geographic condition, climate, and vegetation can influence the genetic structure of lichen populations [14,15,16,17,18,19]. Knowledge of distribution and genetic variations can help better understand the basic ecology of foliicolous lichens as well as the fundamental aspects of the conservation strategy. 

In South Korea, only one genus of foliicolous lichen, *Strigula*, has been recorded from Jeju Island [20]. Jeju Island is located in the southernmost area of South Korea and has the warmest conditions, i.e., a subtropical climate [21]. In addition, the vegetation is different from that of the Korean mainland, which is composed of temperate-subtropical vegetation. This climate and these vegetation conditions allow foliicolous lichens to live on Jeju Island [20]. Recently, a study using molecular phylogeny showed that two novel *Strigula* exist [22]. Among them, *Strigula multiformis* (Mycobank number: MB831530) is the most common species on Jeju Island. This species has been mistaken for *Strigula smaragdula* in South Korea based on their similar morphology, but both the molecular and morphological characteristics of *S. multiformis* can be used to distinguish it from *S. smaragdula* [22]. Although it has distinct characteristics compared to other groups of lichens, the ecology of *Strigula* has been largely unexplored. Given that Jeju Island is the optimal habitat of foliicolous lichens in South Korea, the distribution and the genetic structure of *Strigula* on Jeju Island need to be investigated to better understand the basic biology as well as the conservation practice of foliicolous lichens in South Korea. 

This study examined the distribution and genetic structure of *Strigula* on Jeju Island. In addition, we hypothesized that environmental factors influence the distribution and genetic structures of *Strigula* populations. Specifically, this study had two objectives. First, the distribution of *Strigula* was surveyed on the whole island, and species distribution modeling (SDM) analysis was performed to identify potential habitats and the influence of environmental factors (bioclimatic variables, elevation, and vegetation) on the distribution of *Strigula*. Second, the genetic structure of *S. multiformis* was analyzed using a nuclear ribosomal internal transcribed spacer (ITS) and partial large subunit (LSU) region, and the effects of environmental factors on the genetic structures were analyzed focusing on the geographic distance, climate, and vegetation.

## 2. Materials and Methods 

### 2.1. Sample Collection and Distribution Analysis 

Jeju Island was chosen as a study area due to its distinctive properties as a *Strigula* habitat. Although *Strigula* is generally a pantropical species [2], Jeju Island is located in one of the two coldest areas in Eastern Asia where *Strigula* species are found (i.e., Korea and Japan) [20,23]. It is the closest region to the mainland of South Korea where no foliicolous lichen have been found [20], which suggests that Jeju Island can be a source of a population when *Strigula* disperses to the mainland of the Korean peninsula. In addition, two *Strigula* species on Jeju Island have been newly described in scientific literature [22]. These species may be endemic to South Korea and the knowledge of these species is very limited; thus, the ecology of these species needs to be elucidated before expanding the study to a worldwide scale. Sampling was conducted on Jeju Island during June–July and November in 2018. For describing the distribution of *Strigula*, an island-wide survey for the forests covering the whole island where *Strigula* can live was performed to confirm the previous records and expand the records of *Strigula* habitats. The specimen records from the Korean Lichen Research Center (KoRLI) (2012–2014) and previous studies [20] were used to survey *Strigula* occurrence. The leaves were thoroughly examined to detect *Strigula* (Figure 1), and species were identified morphologically based on previous studies [2,22]. For population genetic analysis, a total of five forests where *S. multiformis* was abundant were chosen to obtain a sufficient number of specimens (Table 1). The leaves with the thalli of *S. multiformis* were collected from a total of six tree stands for each forest.

SDM analysis is the method to predict the distribution and suitable habitat conditions based on species distribution records and environmental variables [24]. Among various algorithms of SDM analysis, maximum entropy modeling (MaxEnt) has shown high performance of model prediction, even with small numbers of occurrence data [25,26,27]; thus, it was used to predict potential habitat distribution of *Strigula* species on Jeju Island. The climate, elevation, and forest availability were used as environmental variables. Nineteen sets of bioclimate data were acquired from the WorldClim database v. 2 [28] as a high-resolution dataset (30 arc-seconds) (Table A1). To avoid overfitting the model due to multicollinearity among climate variables, the variables were reduced using a stepwise backward variable selection procedure based on the variance inflation factor (VIF < 10), using the usdm package [29]. Elevation data were acquired as digital elevation model (DEM) data from the NASA shuttle radar topographic mission (SRTM) dataset [30]. Vegetation distribution and type information (1:5000 resolution) were acquired from the forest geographic information system (FGIS) operated by the Korea Forest Service. The forest availability was filtered based on the forest type and composition to cover the possible host plants of *Strigula* (evergreen broadleaves or mixed forest). Preprocessing of the geographical raster and vector data was conducted using gstudio [31], raster [32], and rgdal [33] packages in R v. 3.5.1 [34], and QGIS v. 3.6 [35]. SDM analysis was performed based on the *Strigula* occurrence records and environmental variables using MaxEnt v 3.3.3 [36], implemented in R package dismo [28] with a maximum of 10,000 background points and a maximum of 500 iterations. The accuracy of the model was evaluated based on the area under the curve (AUC) of the receiving operator curve (ROC). AUC values of more than 0.9, 0.7–0.9, and 0.5–0.7 indicated high, moderate, and low model accuracy, respectively [37,38]. The contribution of an environmental variable to predict the model was measured using the jackknife test. The final model was constructed after removing the low contribution (<1%) variables.

### 2.2. Molecular Experiments

A total of 3–5 thalli from each tree stand were used in the analyses. From each leaf specimen, a single thallus was collected to avoid possible clonal strains living together on the same leaf surface. The genomic DNA was extracted from a thallus of *S. multiformis* using a modified cetyltrimethylammonium bromide (CTAB) method [39]. Since genomic DNA can contain the DNA from other organisms (e.g., host plant, alga, or other fungi), a specific primer (Strig1F, 5-AGSWGCTTAAGATATGGTCG-3) for *S. multiformis* and its close *Strigula* species was developed on the intron site between the ITS5 and ITS1 primer positions [40], using the *Strigula* sequences from GenBank (KF553661-KF553664, MK118870-MK118879). The ITS region was amplified using ITS5/ITS4 or Strig1F/ITS4 [39], and the LSU region was amplified using LROR/LR5 or Strig1F/LR5 [41]. The host plant was identified based on the morphological characteristics and confirmed based on the chloroplast gene for morphological representative samples chosen from each location. The chloroplast maturase K (matK) or ribulose bisphosphate carboxylase large chain (rbcL) was amplified using matK-390F/matK-1326R [42] and rbcL-1F/rbcL-724R [43] primers, respectively. PCR amplification was performed using AccuPower PCR premix (Bioneer, Daejeon, South Korea) under the following conditions: 95 °C for 5 min, 35 cycles of 95 °C for 40 s, 55 °C for 40 s, 70 °C for 1 min, and a final extension at 70 °C for 10 min. Sequencing was conducted using the PCR primer sets on an ABI Prism 3730xl analyzer (Applied Biosystems, Foster city, CA, USA) from Macrogen (Seoul, South Korea). The sequences were checked and edited in MEGA v. 5 [44] and aligned using MAFFT v. 7 [45]. All sequences generated in this study were deposited at GenBank under accession numbers of MN097373-MN097492 for ITS, MN097248-MN097367 for LSU region, and MN103855-MN103866 for the plant chloroplast genes.

### 2.3. Population Genetic Analysis

The genetic variations of *S. multiformis* populations were calculated for the number of variation sites (S), nucleotide diversity (π), number of haplotypes (h), and haplotype diversity (Hd) using DnaSP v. 6 [46]. Prior to analysis, the ITS region was filtered by extracting the partial 18S and LSU region using ITSx v. 1.1 [47]. For the LSU region, the sequence showing a large insertion/deletion (indel) site (58 bp) can result in an over- or underestimation of the genetic diversity of populations. Thus, the sequence of the indel site was imposed based on the most common haplotype (haplotype H01) after adding one nucleotide difference representing an indel event. The haplotype network was generated to determine the relationship between the haplotypes using PopART v. 1.7 [48] based on the TCS method [49]. The demographic history of the population was analyzed using a neutrality test based on Tajima’s D using the DnaSP. A D value near zero indicates neutrality and equilibrium, whereas a significantly high value implies a recent population contraction or balancing selection [50]. 

The genetic differentiation of the populations was examined using a pairwise ϕ_st_ and tested for geographic groups and host plant groups by Analysis of Molecular Variance (AMOVA) using Arlequin v. 3.5 with the option of “compute distance matrix” [51]. The sequences from the same tree stands were grouped as a subpopulation. Discriminant analysis of the principal components (DAPC) was conducted to detect the population structures [52]. This method is similar to STRUCTURE analysis, while it has no assumption of independence between loci. The clustering of individual sequences was performed from k-means clustering, and the number of the cluster was determined based on the Bayesian information criterion (BIC). All analyses associated with DAPC were conducted using the adegenet package [53]. The isolation by distance (IBD) pattern was tested using a simple Mantel test on the vegan package [54] based on geographic distance and genetic distances. Slatkin’s distance (ϕ_st_/(1 – ϕ_st_)) [55] was calculated for the genetic distance. The isolation by environment (IBE) pattern was analyzed for the elevation and Bioclim variables using a simple Mantel test. 

## 3. Results

### 3.1. Distribution and Potential Habitat of *Strigula*

A total of 12 forests were confirmed to have *Strigula* (Figure 2a; Table 1). The bioclimatic variables were selected prior to SDM analysis. Among 19 variables, six variables remained for analysis: BIO01 (Annual mean temperature), BIO03 (Isothermality), BIO07 (Temperature annual range), BIO12 (Annual precipitation), BIO14 (Precipitation of the driest month), and BIO15 (Precipitation seasonality) (Figure A1). In the initial model, BIO1, BIO3, BIO7, BIO12, and BIO15 made a low contribution to predicting the *Strigula* distribution; thus, they were excluded from the final model. A best-fit model containing BIO14, elevation, and forest availability showed high predictive accuracy (AUC = 0.902) (Figure 2b). The forest availability made the highest contribution (67.8%), followed by elevation (20.8%) and BIO14 (11.4%). The predicted suitable habitat existed in evergreen broadleaves or mixed forest (Figure 2a). The total forest area of Jeju Island was calculated to be 62,531 ha, and the area of evergreen broadleaves or mixed forest was 23.3% of the total forest (14,577 ha). In the case of elevation, the low-elevation areas (<500 m) were predicted to be a better habitat. The probability of *Strigula* occurrence decreased with the increasing amount of precipitation in the driest month (BIO14).

### 3.2. Genetic Diversity of *S. multiformis* Populations

A total of 120 thalli of *S. multiformis* collected from five forests on Jeju Island were analyzed using the ITS and LSU regions. The length of alignment and number of segregating sites were 469 bp and 12 for the ITS region, and 906 bp and 15 for the LSU region, respectively. The nucleotide diversity was higher in the ITS regions than in the LSU region. In total, 19 ITS haplotypes and 17 LSU haplotypes were found (Table 2). The most common ITS haplotype was H01 with four populations (17 thalli), followed by H13 belonging to DB and NE populations (31 thalli) (Figure 3a). In the case of the LSU haplotype, haplotype H07, belonging to DB and NE populations (34 thalli), was the most common, followed by H01 with four populations (30 thalli) (Figure 3b). The LSU haplotypes, H02 and H03, had large indel (58 bp) near the LR5 primer site, and they belonged mostly to the AD population except for a thallus from the JJ population. Population-specific haplotypes were detected in 15 haplotypes for the ITS regions (71.4%) and 11 haplotypes for the LSU regions (68.8%), and they comprised 38.3% and 20.8% of thalli, respectively. In the case of the haplotypes shared among multiple populations, one population covered the highest proportions (≥50%). All populations showed no selection and a constant population size by an estimate of Tajima’s D, except for the CJ population (Table 2). The ITS region in the CJ population showed a significantly high Tajima’s D value, which suggests that the selection pressure or recent population had decreased; the LSU region of the CJ population also showed a high value but it was not significant (Table 2).

### 3.3. Genetic Structure of *S. multiformis* Populations

The genetic structure of *S. multiformis* populations was analyzed for the geography and host plant groups using AMOVA (Table 3). For the geographical group, all fixation indices were significant for both the ITS and LSU regions (*p* < 0.01). The high amount of variation was explained by the geographical groups (ITS: 39.7%, LSU: 55.6%), whereas the difference in the thalli within the subpopulation (tree stand) also explained the significant variations (ITS: 47.2%, LSU: 35.2%). The subpopulation difference within the geographical group had a low contribution to the genetic structure (ITS: 13.2%, LSU: 9.2%). For the host plant group, the fixation indices for the different groups were not significant for both the ITS (*p* = 0.548) and LSU (*p* = 0.126). The contribution of the host plant group on explaining the genetic variation was extremely low (ITS: –2.1%, LSU: 5.8%).

A high level of differentiation for geographical populations was detected from the high values of pairwise ϕ_st_ for both the ITS and LSU regions (Table 4). The values of pairwise ϕ_st_ indicated that the population differentiation was high (ITS: 0.148–0.738, LSU: 0.114–0.817) and significant (*p* < 0.05). For the ITS region, the lowest ϕ_st_ value was detected between the Andeok Valley (AD) and Cheonjiyeon (CJ) populations (0.171) and the AD and Jeoji Gotjawal (JJ) populations (0.148). For the LSU region, the Dongbaek hill (DB) and Nabeup forest (NE) populations had the lowest value (0.114). 

DAPC analysis showed that four genetic clusters existed in the total population of *S. multiformis* (Figure 4a). All individuals had a high probability of membership to each cluster (>90%). The composition of genetic clusters was divided geographically into northern (NE and DB) and southern (AD, CJ, and JJ) regions (Figure 4b). For the northern region, cluster 4 was composed of the highest proportions (>56%). Cluster 1 was more common in the southern regions (AD, CJ, and JJ) than in the northern regions. In the southern regions, the proportion of cluster 3 increased along the eastern side from CJ to the JJ population. Cluster 2 was most common in the AD and CJ populations.

The IBD and IBE patterns were analyzed using a Mantel test. The geographic distance did not show a significant relationship with the genetic distances (ITS: r = –0.217, *p* = 0.717, LSU: r = –0.374, *p* = 0.825). The IBE pattern was tested for the elevation and Bioclim variables (BIO1, BIO3, BIO7, BIO12, BIO14, BIO15), whereas no significant relationship was detected between any of the environmental variables and genetic distances. 

## 4. Discussion

The distribution of *Strigula* species was examined and *Strigula* was found only in 12 forests on Jeju Island (Figure 2), which suggests that *Strigula* species are rare and restricted to a small portion of the area on Jeju Island. Lichen distribution is influenced by a range of environmental factors, such as climate [5,56], vegetation [8], host plant physiology [57], soil physiochemistry [58], land use [59], and concentration of pollutants [60,61]. For *Strigula* species, host availability can be one of the most important factors to the establishment of forests because foliicolous lichen can live on the surface of host leaves [2]. *Strigula* are generally found on the leaves of evergreen broadleaf plants [3,62]. In South Korea, *Camellia japonica*, *Litsea* spp., *Machilus* spp., and *Quercus acuta* are known as the host plants of *Strigula* [20,22]; thus, *Strigula* occurs in evergreen broadleaved forests or mixed forests, where the host plant exists. Among the total area of Jeju Island (ca. 180,000 ha), forest covers 34.1% (ca. 62,000 ha) and most of the forest is located on Mt. Halla, where the elevation is high. Considering the forest type and composition, the potential habitat of *Strigula* is only 8.0% of the total area of the island. The SDM analysis supports the importance of host plant presence for *Strigula* distribution. The contribution of the model prediction was highest for the forest availability (Figure 2b). The importance of the forest structure for lichen communities is well known, particularly for epiphyte lichens [8,63,64], and similar characteristics were also found in foliicolous lichen [65].The elevation is the second largest contributing factor to the distribution of *Strigula* (Figure 2b). *Strigula* was found in the lowland forest where the elevation is low (Table 1). On Jeju Island, Mt. Halla is located in the center of the island, and the elevation decreases toward the coast. *Strigula* was not found on Mt. Halla and was only found in lowland forest (6–287 m), which agrees with the previous study that the community of foliicolous lichens is different depending on the elevation and tends to favor lowland areas [23,65]. Precipitation also contributes to the potential distribution of *Strigula* in that *Strigula* occurred in areas with low precipitation in the driest month (Figure 2b). Given the worldwide distribution of *Strigula* in a tropical forest, *Strigula* is expected to favor high precipitation, but the model prediction showed an opposite trend. The mechanism of the precipitation effect on the *Strigula* distribution is unclear and needs to be elucidated. In this study, SDM analysis was performed based on a small number of *Strigula* records from a limited area, which suggests that caution should be exercised when the results from our SDM analysis are interpreted. Although MaxEnt is relatively robust, a small dataset can make biased models and reduce the accuracy of prediction [24,26]. To obtain a more reliable model of *Strigula* distribution, therefore, further study needs to be conducted using an increased number of occurrence records from an expanded study area.

Two of the nuclear ribosomal DNA regions showed genetic variations well within and between *S. multiformis* populations. The ITS and LSU regions are used frequently for phylogenetic and population genetic analysis, and many lichen species showed high variations in these regions [15,17,18,66]. In this study, the ITS and LSU region showed a high level of genetic diversity, which concurs with previous studies [17,66,67,68]. The number of population-specific haplotypes was high in both ITS and LSU regions (Figure 3). In the case of haplotypes shared among populations, they had an uneven proportion biased to a specific population; more than half of thalli in the haplotype were from a single population. Most populations showed a similar level of haplotype diversity, but the CJ population showed low diversity and its Tajima’s D value was significantly high (Table 2), suggesting a demographic history of population contraction. Since the CJ population is located at a tourist site (Cheonjiyeon waterfall), where a thin managed forest exists along the stream, human disturbances may reduce the diversity and population size, as reported for other lichens [64,69]. 

The AMOVA result for the geographic group showed significant differentiation between populations that were also subdivided within individual tree stands (Table 3). In contrast, the host plant did not influence the haplotype composition. This result coincides with the case of epiphytic lichen (*Ramalina menziesii*), which showed a similar population structure among the host plant [11,66]. These results indicate that *S. multiformis* populations on Jeju Island have great diversity within the population and are highly differentiated by geography. Based on DAPC analysis, the genetic structure was divided into the northern and southern areas (Figure 4). This population differentiation, depending on the longitudinal gradient, may be associated with the close geographical distance or similar environmental conditions. On the other hand, the effects of the geographic distance (IBD) and environmental difference (IBE) on the population differentiation were not detected in the *S. multiformis* populations. The IBD pattern has been found depending on the lichen species; *Cetradonia linearis, Parmelina carporrhizans*, and *Parmelina tiliacea* showed IBD patterns [17,18,70]. The different pattern of IBD may be associated with dispersal mechanisms. Generally, sexual propagules (e.g., ascospores) can disperse over long distances, whereas asexual propagules (e.g., isidia, soredia, and conidia) only disperse over a short range [71,72,73,74]. *S. multiformis* can make ascospores, while it is rare for it to produce abundant conidia [22]. Therefore, the long distance between habitats may obstruct the dispersal success due to the capacity of conidia dispersal, which may exceed the scale of the IBD pattern. Environmental variables also did not explain the genetic distance of *S. multiformis* populations, which is different from that of other lichen species. The IBE pattern of climatic variables (e.g., temperature and precipitation) have been detected in *Cetradonia linearis, Nephroma laevigatum*, and *Parmelina tiliacea* [17,19,68]. The IBE pattern was not detected in *S. multiformis* populations possibly because of a lack of relevant environmental variables. In this analysis, the macroclimatic variables from WorldClim were used, but the microclimate (e.g., sampling height or distance from the river) can be important to lichens [19,75]. In addition, the genotype specificity of photobiont can influence the genotype of the lichen populations [76,77,78].

The high genetic differentiation of *S. multiformis* populations may be influenced by the habitat availability. The low level of habitat availability due to forest fragmentation and deforestation can lead to a restriction of gene flow between populations [79,80]. In the center of Jeju Island, forests are highly connected among Mt. Halla. In lowland areas, however, the forests are restricted to a small area and are disconnected [81]. Given that host plant availability is the most important factor affecting *Strigula* distribution (Figure 2b), the disconnected forests in lowland areas mean that *Strigula* experiences a limitation of habitat, which means it is restricted geographically, resulting in a low level of gene flow. Under this condition, the genetic diversity can be reduced, and geography-specific genotypes can disappear easily if the habitat is destroyed.

## 5. Conclusions

The distribution and genetic structure of *Strigula* was examined on Jeju Island. *Strigula* species were rare and distributed in restricted regions. The forest availability made the greatest contribution to the potential niche prediction, followed by elevation and precipitation. *Strigula multiformis* populations had high genetic diversity specific to each location and were highly differentiated by geography. Generally, populations were divided into the northern and southern areas, whereas the factors associated with population differentiation were unclear. Considering the restricted area of the predicted habitats and high differentiation level, *Strigula* on Jeju Island may suffer from habitat limitation and fragmentation. Therefore, the mechanism for the environmental effects on the distribution and genetic structure of *Strigula* needs to be investigated to better understand the ecology of foliicolous lichen and conserve the *Strigula* community in South Korea.

## Figures and Tables

**Figure 1 microorganisms-07-00430-f001:**
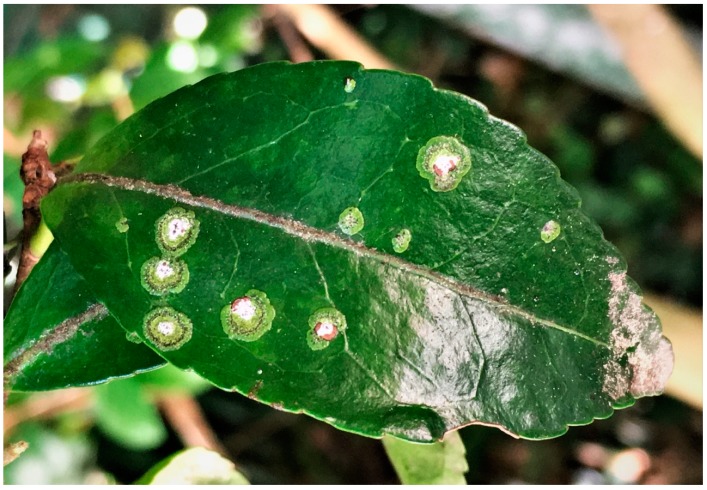
*Strigula multiformis* on the leaf of *Camellia japonica*. Photograph by Jung-Jae Woo.

**Figure 2 microorganisms-07-00430-f002:**
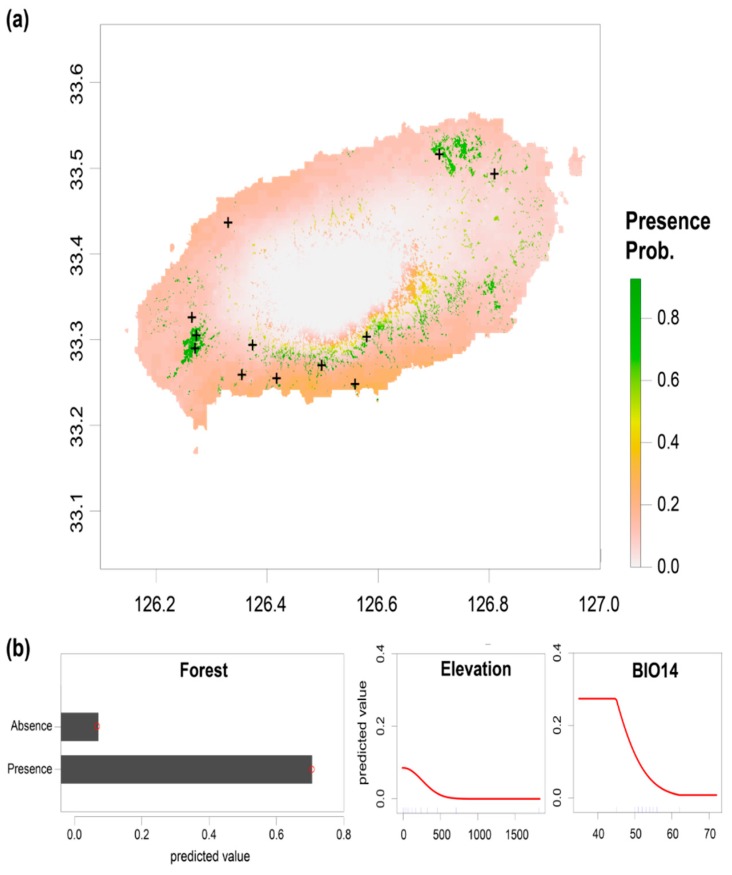
Distribution and potential habitat prediction of *Strigula* on Jeju Island. (**a**) *Strigula* distribution and potential habitats predicted by MaxEnt. The highest probability of its presence is indicated as green. A cross sign indicates the location of *Strigula* occurrence recorded by survey. (**b**) Response plot of environmental variables contributing to the model prediction of species distribution modeling (SDM) analysis.

**Figure 3 microorganisms-07-00430-f003:**
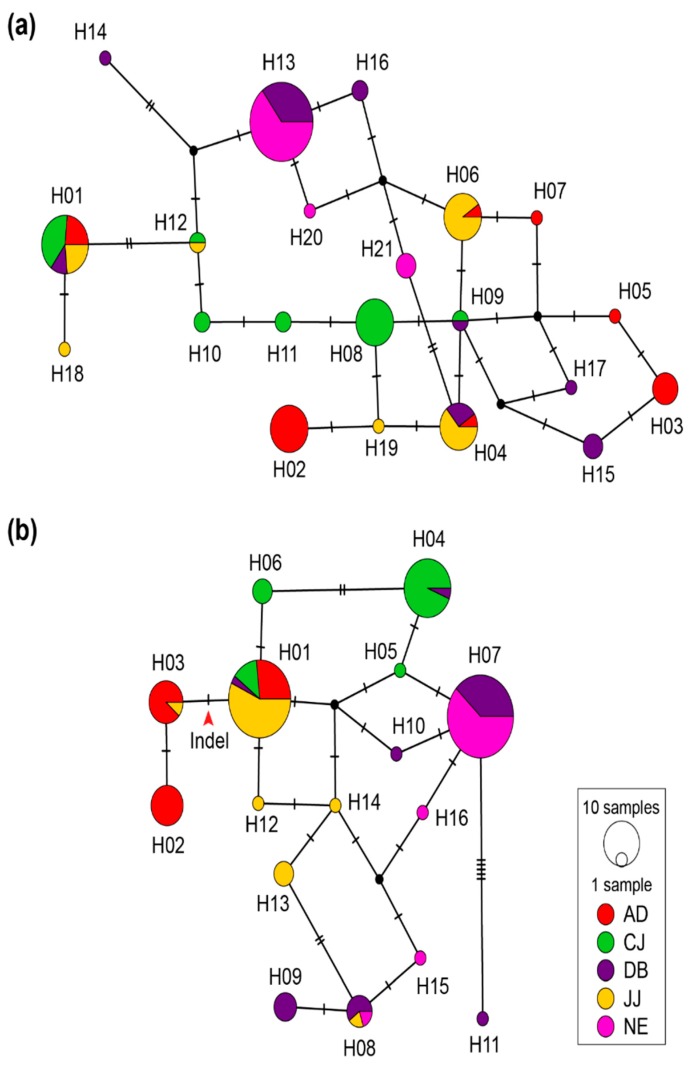
Haplotype networks for (**a**) the ITS and (**b**) the LSU region of the *S. multiformis* populations. The size of the pie chart is proportional to the number of thalli belonging to the haplotype, and the color is according to the geographic populations. The dash on the line represents one mutational step of the haplotype sequence. The red arrow in the LSU network indicates an insertion/deletion (indel) event.

**Figure 4 microorganisms-07-00430-f004:**
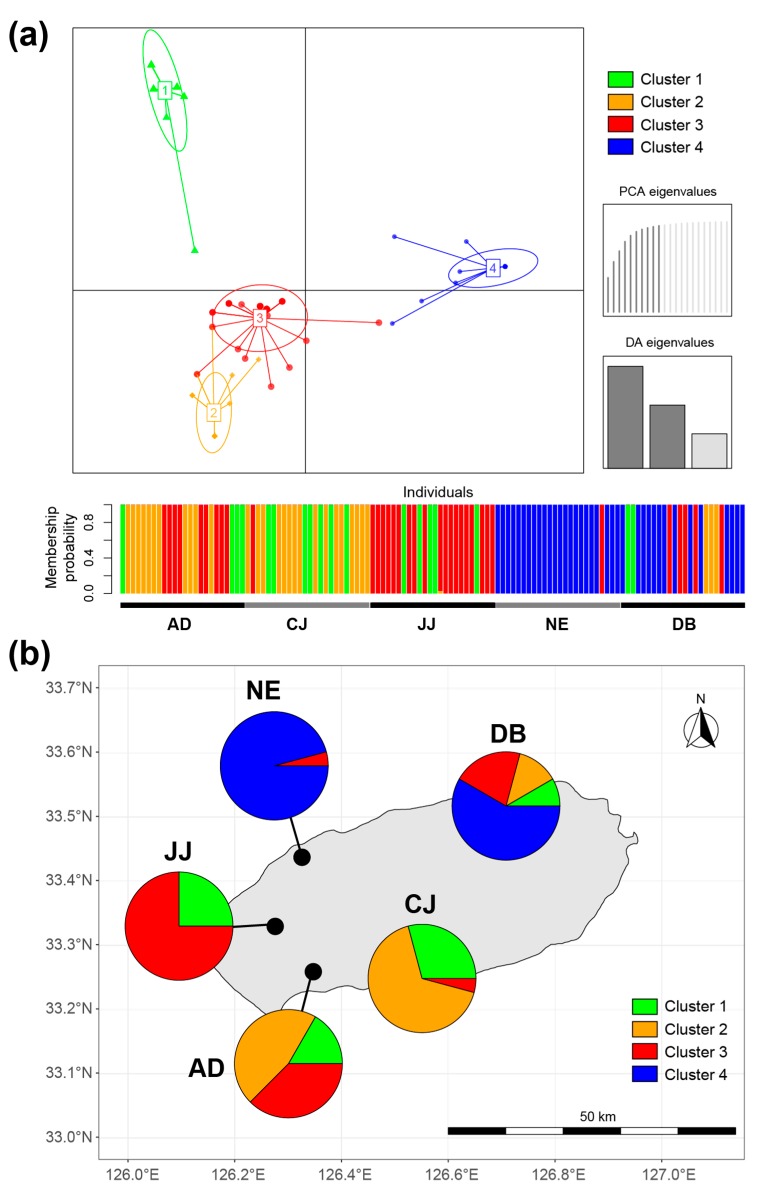
Genetic structure of *S. multiformis* populations based on discriminant analysis of the principal components (DAPC). (**a**) Scatter and membership plot for the DAPC clusters. (**b**) Distribution map of DAPC clusters at each population. The clusters are coded by colors and their size is proportional to the number of thalli (*n* = 25/each population).

**Table 1 microorganisms-07-00430-t001:** List of locations where *Strigula* species have been detected.

No.	Locality (Code)	GPS	Elev. (m)	Species Detected
**1**	**Andeok Valley (AD)** *****	**33°15′23.06′′N, 126°21′16.61′′E**	**33–124**	***S. multiformis***
2	Bija forest	33°29′27.45′′N, 126°48′34.27′′E	131–170	*S. depressa, S. multiformis*
3	Camellia hill	33°17′28.56′′N, 126°22′25.96′′E	262–263	*S. depressa*
4	Cheongsu Gotjawal	33°18′09.29′′N, 126°16′20.93′′E	115–134	*S. multiformis*
5	Cheonjaeyeon	33°15′08.43′′N, 126°25′01.59′′E	45	*S. multiformis*
**6**	**Cheonjiyeon (CJ)**	**33°14′43.93′′N, 126°33′30.58′′E**	**6–32**	***S. multiformis***
**7**	**Dongbaek hill (DB)**	**33°30′49.5′′N, 126°42′36.24′′E**	**77–122**	***S. multiformis***
8	Eongtto waterfall	33°16′03.45′′N, 126°29′53.59′′E	185	*S. depressa*
9	Hwan-Sang Forest	33°19′24.54′′N, 126°15′52.89′′E	132–141	*S. depressa, S. multiformis*
**10**	**Jeoji Gotjawal (JJ)**	**33°17′14.30′′N, 126°16′12.81′′E**	**91–167**	***S. depressa, S. multiformis***
**11**	**Nabeup forest (NE)**	**33°26′02.99′′N, 126°19′46.86′′E**	**89–97**	***S. depressa, S. multiformis***
12	Won-ang Waterfall	33°18′02.47′′N, 126°34′46.11′′E	287	*S. multiformis*

The locations used for population genetic analysis are presented in bold and the population code is provided in parentheses.

**Table 2 microorganisms-07-00430-t002:** Summary of the genetic diversity statistics of internal transcribed spacer (ITS) and large subunit (LSU) regions of *S. multiformis* populations.

Marker	Population	n	S	h	Hd	π	D
ITS	AD	24	9	7	0.743	0.00811	1.39443
	CJ	24	5	6	0.717	0.00509	2.25414 *
	DB	24	10	8	0.772	0.00754	1.06039
	JJ	24	8	6	0.739	0.00557	0.69848
	NE	24	3	3	0.301	0.00159	−0.17843
	Total	120	13	21	0.882	0.00877	1.5262
LSU	AD	24	2	3	0.696	0.00102	1.60933
	CJ	24	3	4	0.533	0.00133	1.26224
	DB	24	12	7	0.685	0.00305	−0.47932
	JJ	24	6	6	0.496	0.00141	−0.62373
	NE	24	4	4	0.239	0.00070	−1.12525
	Total	120	14	16	0.829	0.00310	0.20223

ITS: internal transcribed spacer, LSU: large subunit, n: number of thalii, S: number of variation sites, h: number of haplotypes, Hd: the haplotype diversity, π: nucleotide diversity, D: Tajima’s D (*: *p* < 0.05).

**Table 3 microorganisms-07-00430-t003:** Analysis of molecular variance (AMOVA) results for the ITS and LSU regions of *S. multiformis* populations. The asterisk indicates a significant fixation index (**: *p* < 0.01, ***: *p* < 0.001).

Group	Source of Variance	d.f.	Sum of Squares	Variance Components	% of Variation	Fixation Indices
**ITS**						
Geography	Between groups	4	94.025	0.885	39.7	0.397 ***
	Among subpop.	25	55.442	0.293	13.1	0.218 ***
	Within subpop.	90	94.767	1.053	47.2	0.528 ***
	Total	119	244.233	2.231		

Host	Between groups	2	7.789	−0.043	−2.1	−0.021
	Among subpop.	27	141.677	1.048	51.0	0.499 ***
	Within subpop.	90	94.767	1.053	51.2	0.488 ***
	Total	119	244.233	2.058		
**LSU**						
Geography	Between groups	4	88.917	0.878	55.6	0.556 ***
	Among subpop.	25	28.317	0.145	9.2	0.208 **
	Within subpop.	90	49.933	0.555	35.2	0.648 ***
	Total	119	167.167	1.578		

Host	Between groups	2	12.482	0.086	5.8	0.058
	Among subpop.	27	104.751	0.830	56.5	0.599 ***
	Within subpop.	90	49.933	0.555	37.7	0.623 ***
	Total	119	167.167	1.471		

**Table 4 microorganisms-07-00430-t004:** Pairwise ϕ_st_ for five populations of *S. multiformis*. The results from the ITS and LSU regions are on the lower and upper triangular matrix, respectively. All values are statistically significant (*p* < 0.05).

	AD	CJ	DB	JJ	NE
AD	0	0.667	0.615	0.350	0.817
CJ	0.171	0	0.394	0.523	0.650
DB	0.320	0.386	0	0.468	0.114
JJ	0.148	0.201	0.236	0	0.720
NE	0.648	0.738	0.213	0.641	0

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
