# Peer review of "Distribution of Foliicolous Lichen Strigula and Genetic Structure of S. multiformis on Jeju Island, South Korea"

_microorganisms, 2019, doi:10.3390/microorganisms7100430_

Round 1

Reviewer 1 Report

Although this work is based on scientifically sound experimental approaches, in my opinion it has several flaws that strongly limit its overall quality.
My main remark concerns the selection of the study area, in relation to how the experimental goal was set. As rightly pointed out by the authors, Strigula is a pantropical genus (although many species can also be found in temperate climates). At this point the question is what would be the relevance of the ecological niche of a Korean island in relation to a worldwide distribution of the genus (and / or of some species of this genus).
Although it is perfectly understandable that the authors work in a study area that is close to them, I see many limitations in their proposal of a SDM model based on a relatively limited dataset of presence data and relative to a limited area (if compared with the possible distribution area of the species considered). The risk in these cases is that the resulting model excludes a large fraction of the potential niche of the species (and probably also of the realized one) leading to misleading conclusions.
On the other hand I found a considerable amount of information in the genetic analysis conducted by the authors, although this too was carried out on a small number of populations and despite the fact that the limited extension of the study area can lead to the same typology of errors highlighted in the case of the SDM model.
Although I am aware of the possible logistical problems, I propose some solutions to the authors:
the first is to increase the presence dataset (perhaps with bibliographic data) for the SDM models, extending the study area (at least to the adjacent tropical regions). If this solution would not be feasible, I would suggest giving less weight in the article to the SDM model and to orient the study from a conservation perspective, giving more emphasis to the population genetics.
I regret that I cannot be more positive, but I am sure that with the appropriate modifications the authors have interesting material that can be taken into consideration for a publication.

Author Response

We very appreciated reviewers' comments to improve our manuscript. 

We have revised our MS according to their comments and give our answer as following:

Professional English editing survice will be asked to MDPI after the reviwe process.

Reviewer 1

Although this work is based on scientifically sound experimental approaches, in my opinion it has several flaws that strongly limit its overall quality.

My main remark concerns the selection of the study area, in relation to how the experimental goal was set. As rightly pointed out by the authors, Strigula is a pantropical genus (although many species can also be found in temperate climates). At this point the question is what would be the relevance of the ecological niche of a Korean island in relation to a worldwide distribution of the genus (and / or of some species of this genus).

A: I agree with the reviewer’s comment that the study including the tropical area needs to be performed to understand the world distribution of genus Strigula. In this study, however, we would like to focus on the distribution and genetic structure of Strigula species in Jeju Island, South Korea. The first reason is that Jeju Island is interesting area because it locates two of the coldest area in Eastern Asia where Strigula species are found (e.g. Korea and Japan). In addition, it is the closest region from the mainland of South Korea where no foliicolous lichen has found, which suggests that Jeju Island can be source of populations when Strigula disperses to mainland of Korean peninsula. Second, two Strigula species we studied are newly described to the science recently from Jeju Island. These species can be endemic to South Korea and knowledge of these species is very limited, thus the ecology of these species needs to be elucidated before expanding the study to worldwide scale. We think the results of this study can be one of the steps to understand the ecology and evolution of Strigula species in Eastern Asia and the world.

Although it is perfectly understandable that the authors work in a study area that is close to them, I see many limitations in their proposal of a SDM model based on a relatively limited dataset of presence data and relative to a limited area (if compared with the possible distribution area of the species considered). The risk in these cases is that the resulting model excludes a large fraction of the potential niche of the species (and probably also of the realized one) leading to misleading conclusions.

A: We totally understand what the reviewer concerned for the number of data, but it may be caused by the natural situation of the limited number of Strigula population in Jeju Island. Strigula are rare species in Jeju Island, which makes difficult to add the presence data even we included the previous study and specimen records. Because the small number of data can make bias on the distribution model, we thoroughly choose the method of SDM analysis to MaxEnt. Although it is also influenced by the sample numbers, MaxEnt is relatively robust to small number of samples (Hernandez et al., 2006; Wisz et al., 2008) and it successfully predicted the SDM of rare species that is more or less 10 records (Pearson et al., 2007). For the issue about the study area limited to Korean island, please see the response mentioned above. 

On the other hand I found a considerable amount of information in the genetic analysis conducted by the authors, although this too was carried out on a small number of populations and despite the fact that the limited extension of the study area can lead to the same typology of errors highlighted in the case of the SDM model.

A: We totally understand the reviewer’s concerns. As mentioned above, the small number of Strigula population was due to small population size of Strigula in Jeju Island; only five forests had enough level of Strigula thalli that can be used to population genetic analysis. Although the number of population was small, we think it can be used to overview of the genetic structure of Strigula population in Jeju Island because the size of the island is relatively small and our samples covered whole region of the island. 

Although I am aware of the possible logistical problems, I propose some solutions to the authors:

the first is to increase the presence dataset (perhaps with bibliographic data) for the SDM models, extending the study area (at least to the adjacent tropical regions). If this solution would not be feasible, I would suggest giving less weight in the article to the SDM model and to orient the study from a conservation perspective, giving more emphasis to the population genetics.

I regret that I cannot be more positive, but I am sure that with the appropriate modifications the authors have interesting material that can be taken into consideration for a publication.

A: We would like to thank the reviewer to give thorough suggestions. As the reviewer mentioned, we have accepted second suggestion that reduces the weight on the SDM analysis and conservation perspective in the manuscript: Change of the title, deletion of the paragraph for conservation perspective associated with SDM analysis, and reduction of contents for SDM analysis in Results and Discussion.

Reviewer 2 Report

The manuscript "Species distribution modeling and genetic structure of foliicolous lichen Strigula in South Korea" is very interesting. The methods used are adequate, but the introduction and discussion must be improved and arranged to better highlight the results achieved and the conclusions drawn from them.

Abstract
The abstract lacks specific research results. But firs Authors have to imp
Please remove citations from the abstract.

Introduction
There are no research hypotheses resulting from the two studies objectives.
Although the second aim is rather a description of the method without specifying the purpose for which this molecular analysis was done. The third aim as "in addition" are the results of the identification of the important environmental factors?
I think that this part of manuscript have to improved.

Materials and Methods - Sample collection and species distribution modeling

SDM analysis is not explained. What does it mean "an extensive island-wide survey".

Strigula species diagnosis is missing.

Discussion
Authors have to discussed their aims and hypothesis.
292 AMOVA correct to ANOVA
In the discussion, elements should be ordered so that they relate to the aims set and the corresponding hypothesis.

Author Response

We very appreciated reviewers' comments to improve our manuscript. 

We have revised our MS according to their comments and give our answer as following:

Professional English editing survice will be asked to MDPI after the reviwe process.

Reviewer 2

The manuscript "Species distribution modeling and genetic structure of foliicolous lichen Strigula in South Korea" is very interesting. The methods used are adequate, but the introduction and discussion must be improved and arranged to better highlight the results achieved and the conclusions drawn from them.

Abstract

The abstract lacks specific research results. But firs Authors have to imp

A: The abstract has revised (L15-24).

Please remove citations from the abstract.

A: There is no citation in the abstract.

Introduction

There are no research hypotheses resulting from the two studies objectives.

A: The hypothesis was added (L57-59).

Although the second aim is rather a description of the method without specifying the purpose for which this molecular analysis was done. The third aim as "in addition" are the results of the identification of the important environmental factors? I think that this part of manuscript have to improved.

A: We have revised the paragraph more clearly (L57-65).

Materials and Methods - Sample collection and species distribution modeling

SDM analysis is not explained.

A: We added the background information of SDM analysis (L82-86).

What does it mean "an extensive island-wide survey".

A: This is a survey for the forests covering the whole island where Strigula can live. We changed the word and added the detail explanation (L69-71).

Strigula species diagnosis is missing.

A: We added the procedure of Strigula identification (L72-74).

Discussion 

Authors have to discussed their aims and hypothesis.

A: We have revised Discussion more clearly (L241-266).

292 AMOVA correct to ANOVA

A: AMOVA is not ANOVA but an Analysis of Molecular Variance.

In the discussion, elements should be ordered so that they relate to the aims set and the corresponding hypothesis.

A: We have ordered Discussion from the Strigula distribution to population genetic structures.

Round 2

Reviewer 1 Report

The response of authors to my previous concerns is accurate and clarify the limit of application and interpretation of their results sufficiently. Accordingly, the authors revised the manuscript following my remarks.

Also, some additional references in the revised version of the paper (e.g., work by Edith) help the reader to identify more clearly the potential limits of applications of the methods used, allowing contextualizing the outcomes from a broader perspective.

I appreciate that an extension of the dataset on which the SDM analysis was performed is not yet feasible, due to the limited knowledge on the distribution of the studied species. Nevertheless, I agree with the authors that their work gives an original insight into the ecology of follicolous Strigula and can contribute to designing active conservation measures in the next future. I encourage the authors to follow this topic, possibly focusing on population-based works which may elucidate the patterns of distribution of the species at a local and landscape scale.

Based on these considerations, I feel that the manuscript by Oh and collaborators is now suitable for publication in Microorganisms.

Author Response

Thank you very much for your kind decision of the revised MS as "accepted".

Reviewer 2 Report

The authors have sufficiently corrected the text according to my comments.

Author Response

(The authors gave the same response as above.)
